In vitro fermentation characteristics of polysaccharide from Scrophularia ningpoensis and its effects on type 2 diabetes mellitus gut microbiota

Zhao Yang 1
Wen Juwei 1
Yang Yu 1
Jia Lina 2
Ma Qian 2
Jia Weiguo 3
Qi Wei 1 qiweismiling@126.com
1 College of Life Science, Zhuhai College of Science and Technology , Zhuhai , China
2 College of Biotechnology, Tianjin University of Science and Technology , Tianjin , China
3 The Center of Gerontology and Geriatrics, National Clinical Research Center of Geriatrics, West China Hospital, Sichuan University , Chengdu , China
Brygadyrenko Viktor
Electronic publication date: 2025 May 5
Publication date: 2025
Volume: 13
Electronic Location ID: e19374
Received 2024 Dec 5; Accepted 2025 Apr 7
Copyright: © 2025 Zhao et al.
Copyright year: 2025
Copyright holder: Zhao et al.
License: This is an open access article distributed under the terms of the Creative Commons Attribution License, which permits unrestricted use, distribution, reproduction and adaptation in any medium and for any purpose provided that it is properly attributed. For attribution, the original author(s), title, publication source (PeerJ) and either DOI or URL of the article must be cited.
License URL: https://creativecommons.org/licenses/by/4.0/

Keywords: Polysaccharide, T2DM, Gut microbiota, Scrophularia ningpoensis

Funding: Characteristic Innovation Projects of Ordinary Universities in Guangdong province of China 2024KTSCX015 Zhuhai College of Science and Technology “Three Levels” Talent Construction Project of China This work was supported by the characteristic innovation projects of ordinary universities in Guangdong province of China (2024KTSCX015), and the Zhuhai College of Science and Technology “Three Levels” Talent Construction Project of China. The funders had no role in study design, data collection and analysis, decision to publish, or preparation of the manuscript.

==============================
Background

Increasing evidence has shown a close relation between the pathogenesis of type 2 diabetes mellitus (T2DM), which is a global health problem with multifactorial etiopathogenesis, and gut microbiota.

Methods

During in-vitro fermentation of Scrophularia ningpoensis (known as Xuanshen) polysaccharide (SNP) by T2DM gut microbiota, effects of SNP on the gas content, production of short-chain fatty acids (SCFAs), metabolite profile and microbiota composition were studied.

Results

Analysis of chemical compositions indicates that the total sugar content of SNP was found to be as high as 87.35 ± 0.13% (w/w). SNP treatment significantly improved the gas volume and composition in T2DM fecal matter. Moreover, intestinal flora degraded SNP to produce SCFAs, thus regulating SCFA production and composition. Metabolomic analysis implied that SNP shows potential to regulate the five gut metabolites (L-valine, L-leucine, L-isoleucine, L-alanine, and xylitol) in T2DM fecal matter. Furthermore, dysbiosis of gut microbiota induced by T2DM was reversed by SNP. The evidence includes decreasing Firmicutes/Bacteroidota ratio at phylum level promoting proliferation of the bacterial abundance of Dorea, Parabacteroides, Faecalibacterium, and Lachnospira and decreased bacterial abundance of Escherichia—Shigella. Based on these findings, the action mechanism of SNP against T2DM was clarified by reshaping microbiota and regulating intestinal metabolites, and a novel target was provided for interventions of T2DM.

Introduction

Human health has always been a concern; this is particularly the case as rates of incidence of various diseases and metabolic disorders increase. Type 2 diabetes mellitus (T2DM) is a chronic metabolic disease featuring hyperglycemia and is generally accompanied by dysfunctions of fat, protein, and carbohydrate metabolism. The diabetic population is expected to grow from 578.4 million in 2030 to 700.2 million by 2045, 90% of which are due to T2DM. T2DM has substantially threatened global human health (DeFronzo et al., 2015; Jiang et al., 2022; Que et al., 2021).

Evidence shows the involvement of intestinal microbiota of human in lots of health conditions and diseases (Chen et al., 2021). The gastrointestinal tract is a complex ecosystem comprising over 1,000 distinct species of microbiota. Changes within this ecosystem, including shifts in microbial community composition, increases in pathogenic microorganisms and alterations in microbial metabolite profiles, can disrupt host homeostasis. Such disruptions may contribute to the development of various diseases (Chen et al., 2024). T2DM was the first disease reported in association with gut microbiota in 2010 (Larsen et al., 2010). Existing research into T2DM onset and progression has identified intestinal floral imbalance as a critical factor for T2DM patients, their gut microbiota contained rich gram-negative bacteria, particularly those in phyla Bacteroidetes and Proteobacteria (Dash et al., 2023). Outer membranes of these bacteria contain lipopolysaccharide (LPS) that may cause metabolic endotoxemia, which has a meaningful relation to macrophage-elements, oxidative stress, and setup inflammatory cytokines that induce insulin resistance (IR) (Cunningham, Stephens & Harris, 2021). Thus, implementation of targeted interventions on the intestinal microbiota can efficiently reshape the equilibrium of the gut microbiotal structure and function, as well as its metabolites. These interventions can also modulate the multifaceted interactions between the intestinal barrier and the microbiota, resulting in a significant improvement in T2DM (Chen et al., 2024). Additionally, the enhancement of insulin sensitivity through changing the composition of gut microbiota has attracted interest among numerous researchers. This finding indicates that targeted interventions designed to modulate the gut microbiota could serve as a potentially effective strategy for managing T2DM (Liu et al., 2022a).

Currently, T2DM drugs mainly include sulfonylureas, biguanides, peroxisome proliferator-activated receptor-γ (PPARγ) agonists, α-glucosidase inhibitors, incretin mimetics, and SGLT2 inhibitors (Padhi, Nayak & Behera, 2020; Sairam et al., 2024). Sulfonylureas enhance the release of insulin from pancreatic islets; biguanides decrease the production of hepatic glucose; PPARγ agonists enhance insulin action; α-glucosidase inhibitors interfere with glucose absorption in gut. The primary mechanism of incretin mimetics is to modulate the incretin system. They bind to and activate glucagon-like peptide-1 (GLP-1) receptors located on pancreatic β-cells, thereby triggering insulin synthesis and secretion. SGLT2 inhibitors promote glycosuria by suppressing the reabsorption of glucose in the kidneys (Hansen, Vilsbøll & Knop, 2010). At therapeutic doses, the SGLT2 inhibitors lead to the excretion of approximately 60–100 g of glucose, which directly reduces glucose levels in systemic circulation and contributes to lowering the level of blood glucose (Brown et al., 2021). Nevertheless, their efficacy is far from satisfactory owing to drug resistance, their high price, and side effects, including severe hypoglycemia, low therapeutic efficacy caused by ineffective or improper dosage regimens, weight gain, low potency, and changed adverse effects because of lack of target specificity and drug metabolism, and problems pertaining to permeability and solubility (Han et al., 2022; Sairam et al., 2024; Wang et al., 2016). To address these problems, many efforts have been made to find natural products that are efficient in alleviating T2DM with no severe adverse effects (Fei et al., 2024). To prevent and treat hyperglycemia, researchers have focused increasing efforts on the discovery of active components in medicinal plants, particularly from traditional Chinese medicines (TCM) (Wang et al., 2023). Polysaccharides from natural sources, as macromolecular carbohydrates, cannot be digested and absorbed in human intestine, so they can reach the colon to promote growth of selective probiotics. They show a beneficial effect on the metabolism of glycolipid. Active polysaccharides, as a natural microecological regulator for intestine, have been extensively used to regulate intestinal tumors, microecology, and inflammation considering the stable efficacy, rich availability, and slight side effects (Fang et al., 2019). Previous research proved that polysaccharides derived from plants are regulatory factors beneficial for intestinal flora disorder in T2DM patients (Kakar et al., 2021; Liu et al., 2024b; Wu et al., 2019c).

Scrophularia ningpoensis Hemsl is a famed popular TCM and health food commonly known as Xuanshen. It contains abundant nutrients, like alkaloids, amino acids, and flavonoids, particularly carbohydrates, which are key effective agents in S. ningpoensis (Gong et al., 2020; Hua, Qi & Yu, 2014). Prior research has found that the aqueous extract of S. ningpoensis reduces fasting blood glucose level and increases the insulin content in T2DM rats (Lu et al., 2017). This extract improves insulin sensitivity by modulating the AMP-activated protein kinase (AMPK)-mediated suppression of the NLRP3 inflammasome and alleviates pancreatic β-cell pyroptosis via AMPK-dependent inhibition of NLRP3/GSDMD activation (Yan et al., 2022). Polysaccharides derived from S. ningpoensis exhibit hypoglycemic properties, enhancig carbohydrate and fat metabolism, increasing insulin levels, and significantly reducing blood glucose. They also inhibit the overproduction of inflammatory cytokines and modulate the MAPK pathway, including ERK, JNK and p38 proteins (Ma et al., 2019; Ren et al., 2021). Many polysaccharides, serving as macromolecular prebiotics, have been shown to promote the proliferation of intestinal probiotics and improve host metabolic functions (Wang et al., 2016). However, limited research is available on the influences of SNP on gut microbiota composition and metabolism in T2DM patients. In-vitro models provide the advantage of reducing influences from the complex in-vivo environment (Wu et al., 2023). An in-vitro anaerobic fermentation model of SNP was established by taking fresh feces of eight T2DM patients and eight healthy humans to explore how SNP affects gas production and microbial communities in the intestine. Besides, variations of gut microbial metabolites and intestinal microbiota were jointly explored using a metabolomic method. The present research estimated the effect of SNP treatment upon T2DM and studied the underlying mechanisms in which intestinal flora and metabolites are involved, offering experimental data for developing SNP healthcare drugs or products to prevent and treat T2DM.

Materials and Methods

Reagents

S. ningpoensis was purchased from Yunnan Province, China. Acetic acid, propionic acid, butyric acid, isobutyric acid, valeric acid, isovaleric acid, trans-2-butenoic acid, ribitol, and N-methyl-N-trimethylsilyl-trifluoroacetamide (MSTFA) were bought from Sigma-Aldrich Inc. (St. Louis, MO, USA). Yeast extract, Casitone, and fatty acid (YCFA) medium were kindly provided by Hangzhou Hailu Biotechnology Co., Ltd (Hangzhou, China). Other chemical reagents were all ordered from Solarbio (Beijing, China).

Preparation of SNP

Crude SNP was extracted using a method from previous research, albeit with certain modifications (Chen et al., 2021). S. ningpoensis powder was mixed with deionized water (1:8, w/v) and heated to 90 °C for 7 h. After centrifuging the extract at 6,000 rpm for 10 min, the supernatant underwent deproteinization by using the Sevag method and precipitated using ethanol (95%, v/v). Thereafter, the precipitate was lyophilized to obtain crude SNP.

Determination of physico-chemical characteristics of SNP

The method of phenol-sulfuric acid (Dubois et al., 1951) and bovine serum albumin (BSA) as standard (Chen et al., 2018) were separately used to estimate total carbohydrate and protein contents of SNP.

Simulated gastric and small intestinal digestions of SNP were analyzed with reference to previous approaches (Minekus et al., 2014). In short, preparation of simulated gastric buffer followed Table S1. The simulated gastric fluid was prepared by mixing 330 μL pepsin (3,000 U/mL), 4 mL simulated gastric buffer, and 25 μL CaCl2 (0.3 M). For simulated gastric digestion, after mixing a polysaccharide solution of the same volume with the simulated gastric buffer, the mixture was incubated in a water bath at 37 °C, while maintaining the reaction solution at pH 3. Two hours after that, samples were taken, followed by inactivation at 100 °C for 5 min. Preparation of simulated small intestinal buffer followed the procedure outlined in Table S1. This simulated intestinal fluid comprised 4 mL small intestinal buffer, 100 μL CaCl2 (0.3 M), 250 μL pancreatin (4,000 U/mL), 400 μL bile salt (100 mg/mL), and 5 mL simulated gastric digestion. The final step was 2 h of digestion of the mixed solution in a water bath at 37 °C, and then enzyme inactivation was made at 100 °C for 5 min.

Inclusion and exclusion criteria for participants

A diabetes diagnosis can be determined by the presence of diabetes-related symptoms such as polydipsia, polyuria and unexplained weight loss, along with one of the following diagnostic criteria: fasting plasma glucose levels of ≥7.0 mmol/L (whole blood of ≥6.1 mmol/L), glycated hemoglobin (HbA1c) of ≥6.5%, random venous plasma glucose of ≥11.1 mmol/L, or a 2-h plasma glucose concentration of ≥11.1 mmol/L, following a 75-g anhydrous glucose oral glucose tolerance test (OGTT) (Society, 2024; World Health Organization and International Diabetes Federation, 2006).

The following inclusion criteria apply: 18–80 years old, the body mass index (BMI) of 18–35 and diagnosis of T2DM at least 6 months before the study.

The following exclusion criteria apply: antibiotic treatments in the 3 months before the research, pregnancy, or lactation.

Experimental design of in-vitro fermentation

The enrolment of 16 human volunteers was supervised, and the research was approved, by the Biomedical Ethics Committee of West China Hospital of Sichuan University, under protocol number 2018 (286). Written informed consent to participate in the study was obtained from volunteers. Eight females and eight males were included. The in-vitro fermentation was the same as that by Liu et al. (2020). The 10% (w/v) stool suspension was formed by collected fresh stool samples and homogenized then using 0.1 M phosphate buffer (pH 7.0). After adding 1 mL fecal suspension, the 9 mL YCFA media containing or not containing SNP (50.0 mg) were transferred to a BSP-100 anaerobic box (Shanghai Boxun Industrial Co., Ltd, Shanghai, China) to be incubated at 37 °C for 24 h. Four treatments were used in this study: (1) original healthy control (HC); (2) original T2DM control (T2DM); (3) SNP group of fecal slurry from HC (HC+SNP); and (4) SNP group of fecal slurry from T2DM (T2DM+SNP).

16s rRNA sequencing microbial composition

The fermentation sample was centrifuged, followed by extraction of microbial genomic DNA through use of a Stool Genomic DNA Extraction Kit (Solarbio, Beijing, China) following the manufacturer protocols. Primer pairs 338 F (5′-ACTCCTACGGGAGGCAGCAG-3′) and 806R (5′-GGACTACHVGGGTWTCTAAT-3′) were adopted to amplify hypervariable regions V3–V4 of the 16S rRNA gene of bacteria using an ABI GeneAmp® 9700 PCR thermocycler (ABI, CA, USA). Majorbio Biotechnology Co., Ltd (Shanghai, China) was entrusted to perform 16S rRNA sequencing. Quality control was followed by grouping of selected sequences into operational taxonomic units (OTUs) through UPARSE (version 10.1 http://drive5.com/uparse/). OTUs were taxonomically annotated using Ribosomal Database Project (RDP). The microbiota compositions and diversity, as well as correlations of gas, short chain fatty acids (SCFAs) and significantly different metabolites were analyzed.

Measurement of the gas volume and composition

The media were collected after in-vitro fermentation for 24 h. A gas detector (Shanghai Yijie Industrial Safety Equipment Co., Shanghai, China) was used to measure the total gas volume in fermentation flasks, and the contents of methane (CH4), carbon dioxide (CO2), hydrogen (H2) and hydrogen sulfide (H2S).

Quantitative analysis of SCFAs

The gas chromatograph (GC, Shimadzu, GC-2010 Plus, Kyoto, Japan) equipped with a DB-FFAP column (Agilent Technologies, Santa Clara, CA, USA) and an H2 flame ionization detector was adopted to measure the concentrations of SCFAs, such as acetic acid, propionic acid, butyric acid, isobutyric acid, valeric acid, and isovaleric acid in culture filtrates. The internal standard was trans-2-butenoic acid (Bai et al., 2017).

Quantification of fecal metabolites via gas chromatography–mass spectrometry

The fermentation samples were centrifuged 12,000g for 10 min at 4 °C. Subsequently, 200 μL of the supernatant was harvested, and ribitol was added as the internal standard at a concentration of 0.6 μg/mL. After using a reported approach (Qi et al., 2022) for oximation and silylation derivatization, the metabolites of the samples were analyzed by an Agilent 7890B series gas chromatography (GC) system with an HP-5 column (30 × 0.25 µm × 0.25 mm) coupled to an Agilent 5977B quadrupole mass. The oven was programmed to have an initial temperature of 50 °C, a holding time of 2 min, a rapid temperature rise to 270 °C at 5 °C/min, another rise to 290 °C at 2.5 °C/min, a final rise to 310 °C at 10°C/min and a holding time of 4 min. The detector and injector temperatures were set to 250 °C, and the carrier gas was helium, which flowed at a rate of 1 mL/min. For mass spectrometry (MS) acquisition, the electron-impact mode was used to scan from 50–700 m/z according to programming. Each compound was identified by comparison with the NIST05a library.

Statistical analysis

Data analysis was performed using SPSS 22 (IBM, New York, NY, USA), with results shown as means ± standard deviations (SDs) (n = 8). GraphPad Prism 9 (GraphPad Software Inc., San Diego, CA, USA) was used for drawing the gas and SCFAs. The Majorbio Cloud Plat-form (http://www.majorbio.com) was adopted for analysis of 16 s rRNA sequencing data. Metabolic data were analyzed using Metaboanalyst 5.0 (https://metaboanalyst.ca) and SIMCA (Sartorius, Germany). The KEGG database was used for studying the metabolic pathways. Spearman correlation analysis was performed to ascertain the correlations of fecal microbiota with various metabolites.

Results

Physico-chemical characterization of SNP

The yields of crude polysaccharide and its total sugar content, protein content were determined. The yield of crude polysaccharide of 1.08 ± 0.09% was attained from water-extracted S. ningpoensis, indicating the presence of the relatively high polysaccharide content (Tong et al., 2021). The total sugar and protein contents were 87.35 ± 0.13% (w/w) and 2.19 ± 0.11% (w/w), respectively.

Changes in reducing sugar and total sugar during simulated digestion of SNP in-vitro are listed in Table 1. During the process, the total sugar content of polysaccharide remained relatively stable while the reducing sugar content exhibited a mild increment (P > 0.05). Polymers are readily formed by polysaccharide in the aqueous solutions. In the acidic environment of the stomach, polymers may be destructed and glycosylic bonds are broken. This indicates that SNP can barely be digested or absorbed by the stomach and small intestine, which enables its arrival at the colon.

Table 1 Total sugar and reducing sugar contents of SNP during in-vitro digestion.

	Total sugar (mg/mL)	Reducing sugars (mg/mL)	
Gastric juice			
0 h	0.2109 ± 0.0023	0.0392 ± 0.0024	
2 h	0.2114 ± 0.0012	0.0418 ± 0.0092	
Small intestinal			
0 h	0.2335 ± 0.0445	0.0602 ± 0.0027	
2 h	0.2373 ± 0.0141	0.0722 ± 0.0043	

Effect of SNP treatment on microbiota composition in T2DM human fecal matter

Figure 1 displays the alpha diversity, which reflects the community diversity and abundance of samples. Generally, abundance coverage estimates (ACE) and Chao indices were influenced by species abundance while Simpson and Shannon indices were under effects of community diversity (Kong et al., 2021). As shown in Fig. 1A, the results showed that Chao and ACE indices did not differ significantly, suggesting the absence of significant difference in community abundance among groups. Whereas, the Shannon index in T2DM+SNP group and HC+SNP group exhibited a significant decrease (P < 0.05), suggesting indicating a lesser level of community diversity in these groups.

Figure 1 Effects of SNP upon the diversity of fecal microflora from humans.

(A) ACE, Chao, Shannon, and Simpson indices; Comparison of bacterial compositions across all groups based on (B) PCoA and (C) PLS-DA at the OTU level. Data are presented as means ± SDs (n = 8).

β diversity, also termed between-habitat diversity, is generally used for diversity comparison across ecosystems. Besides, the research also evaluated principal coordinate analysis (PCoA), which reveals diversity in the compositions of gut microbiota. Figure 1B shows that the four groups have a notable clustering of compositions of gut microbiota, with PC1 (28.61%) and PC2 (15.15%) exhibiting a total variance of 43.76%. Conforming to PCoA results, a similar trend was also shown in the PLS-DA results (Fig. 1C). These results further showed the ability of SNP to modulate the structure and richness of intestinal flora.

Figures 2 and 3 displays evaluation results of the structure of microbial communities at phylum and genus levels. Microbes basically comprised Firmicutes, Proteobacteria, Bacteroidetes, Actinobacteria, and Desulfobacterota at the phylum level (Fig. 2). Compared with the HC, the Firmicutes/Bacteroidetes (F/B) ratio of T2DM is reduced, showing similarity to results in previous research (Fassatoui et al., 2019; Hamasaki-Matos et al., 2021). Moreover, the F/B ratio was reversed by T2DM+SNP group statistically (vs. T2DM group, P < 0.05). The F/B relative ratio in the host serves as an effective indicator for whether intestinal homeostasis is favorable or not (Stojanov, Berlec & Štrukelj, 2020). According to previous research, Bacteroidetes and Firmicutes contain intestinal bacteria in nine genera, including Phascolarctobacterium, Faecalibacterium, Blautia, and Bacteroides, and the both phyla generate SCFAs (Li, Hu & Xiong, 2023). As a major phylum of gram-negative bacteria, Proteobacteria contains many pathogenic microorganisms, including Shigella, Salmonella, and Escherichia coli. These may induce mild inflammations and even chronic colitis in severe cases (Zhang et al., 2022b). Bacteroidetes can cause severe illness; Bacteroides fragilis involves in the majority of anaerobic infection (Yekani et al., 2020). As shown in Figs. 3, 4, abundances of Escherichia-Shigella, Lactobacillus, Parabacteroides, Bifidobacteriumin, and Enterococcus in the T2DM group increased at the genus level, and those of Eggerthella, Phascolarctobacterium, Desulfovibrionaceae, Bilophila, Dorea, Butyricicoccus, Lachnospiraceae, Faecalibacterium, Lachnoclostridium, and Lachnospira decreased (vs. HC group, P < 0.05). Moreover, the T2DM+SNP group showed statistical reverse in abundances of Dorea, Escherichia-Shigella, Parabacteroides, Faecalibacterium, and Lachnospira (vs. T2DM group, P < 0.05).

Figure 2 Bacterial taxonomic profiles at phylum levels.

Figure 3 Bacterial taxonomic profiles at genus levels.

Figure 4 Relative abundance of genera with a significant difference.

The importance of comparisons between indicated groups was evaluated by conducting Kruskal-Wallis tests on the significant difference, *P < 0.05, **P < 0.01, and ***P < 0.001.

Moreover, specific bacterial populations across groups were evaluated through effect size analysis (LEfSe, Fig. 5) and linear discriminant analysis (LDA, Fig. 6). Figure 6 shows that the HC, T2DM, HC+SNP, and T2DM+SNP groups were found to contain 26, 20, 10, and 5 significantly different OTUs (LDA score > 2.5). Phascolarctobacterium, g_unclassified_f_Lachnospiraceae Dorea, Bilophila, Eggerthella, Lachnoclostridium, Candidatus_Soleaferrea, Desulfovibrionaceae, Holdemania, Butyricicoccus, Coprococcus, and Clostridium_innocuum were dominant genera in the HC group. Dominant genera in the T2DM group included Escherichia-Shigella, Parabacteroides, Enterococcus, Halomonas, and Flavobacterium. Dominant genera in HC+SNP group comprised Lactococcus, Faecalibacterium, g_norank_f_Lachnospiraceae, Bacteroidales, and Lachnospira. The T2DM+SNP group was dominated by genus Bifidobacterium.

Figure 5 Significant differences of microbiota across various groups were identified using LEfSe.

Figure 6 LDA scores of enriched bacterial taxa (LDA > 2.5 of LEfSe).

The further to assess the associations between changed gut microbiota in human fecal samples and SNP, microbiota were selected based on both up-regulation or down-regulation in the T2DM+SNP group (vs. T2DM group) and HC+SNP group (vs. HC group) (Table 2). The results showed that Lactobacillus, Faecalibacterium, Bifidobacterium, and Lachnospira were up-regulated after treatment of SNP. While Escherichia-Shigella, Eggerthella, Phascolarctobacterium, Desulfovibrionaceae, Bilophila, Dorea, Parabacteroides, Butyricicoccus, and Lachnospiraceae were down-regulated after treatment of SNP. These results indicated that it is possible for SNP to modulate dysbiosis of gut microbiota.

Table 2 Species abundance in different treatment groups.

Species name	HC group (%)	HC+SNP group (%)	T2DM group (%)	T2DM+SNP group (%)	
Lactobacillus	0.001 ± 0.003	0.36 ± 0.54	1.32 ± 1.78	3.19 ± 4.51	
Faecalibacterium	0.25 ± 0.33	2.02 ± 2.68	0.04 ± 0.06	1.26 ± 3.21	
Bifidobacterium	2.87 ± 3.00	13.75 ± 13.86	3.42 ± 4.73	18.30 ± 22.95	
Lachnospira	0.09 ± 0.11	0.13 ± 0.15	0.03 ± 0.05	0.07 ± 0.13	
Escherichia-Shigella	35.94 ± 16.54	8.15 ± 12.39	48.89 ± 18.43	13.54 ± 22.60	
Eggerthella	0.51 ± 0.83	0.02 ± 0.02	0.25 ± 0.18	0.01 ± 0.01	
Phascolarctobacterium	7.42 ± 8.51	1.54 ± 2.10	5.09 ± 5.45	0.29 ± 0.54	
Desulfovibrionaceae	0.18 ± 0.18	0.001 ± 0.003	0.09 ± 0.12	0.01 ± 0.01	
Bilophila	0.95 ± 1.36	0.03 ± 0.03	0.35 ± 0.34	0.02 ± 0.02	
Dorea	1.80 ± 1.60	0.63 ± 1.19	0.76 ± 0.90	0.20 ± 0.41	
Parabacteroides	1.49 ± 1.37	0.42 ± 0.36	3.21 ± 4.41	0.30 ± 0.31	
Butyricicoccus	0.11 ± 0.14	0.09 ± 0.08	0.02 ± 0.06	0.01 ± 0.02	
Lachnospiraceae	2.21 ± 3.00	0.37 ± 0.36	0.21 ± 0.43	0.07 ± 0.16	

Effect of SNP treatment in-vitro fermentation on the gas volume and composition in T2DM human fecal matter

Total gas volume, CO2, CH4, H2, and H2S are displayed in Fig. 7. Total gas volume, CO2, CH4, H2, and H2S decreased by 32.44%, 25.55%, 89.44%, 30.67%, and 64.54% in the T2DM group compared with those in the HC group. While total gas volume, CO2, CH4, H2, and H2S showed increases of 119.32%, 477.01%, 3210.52%, 5.51%, 9.13% in the T2DM+ SNP group compared with those in the T2DM group; there were no differences for total gas volume, CO2, CH4, H2, and H2S between T2DM+ SNP and HC+ SNP groups. The result showed that the presence of SNP could contribute to the regulation of the gas volume and composition in T2DM fecal, with a low possibility of production of H2 and H2S and a high possibility of production of CO2 and CH4.

Figure 7 Gas production, methane (CH4), carbon dioxide (CO2), hydrogen (H2) and hydrogen sulfide (H2S) after 24-h in-vitro fermentation with T2DM human fecal Microflora.

One-way ANOVA was used to analyze results expressed as means ± SDs, and then post-hoc Tukey’s multiple comparisons tests were performed. *P < 0.05 and **P < 0.01.

Effect of SNP treatment on SCFAs production and composition in T2DM human fecal matter

Figure 8 shows that the T2DM group has lower contents of total SCFAs, acetic acid, propionic acid, butyric acid, isobutyric acid, valeric acid, and isovaleric acid, when compared with the HC group (P < 0.05). While contents of total SCFAs, acetic acid, propionic acid, butyric acid, isobutyric acid, and valeric acid in T2DM+SNP increased by 51.07%, 20.40%, 120.82%, 67.40%, 30.88%, and 315.04% compared with the T2DM group. Moreover, the T2DM+SNP and HC+SNP groups were similar in terms of the production of total SCFAs (acetic acid, propionic acid, butyric acid, valeric acid, isobutyric acid, and isovaleric acid). This suggested the degradation of SNP by intestinal flora to produce SCFAs and the regulation of SCFA production and composition in human fecal matter.

Figure 8 Total SCFAs (acetic acid, butyric acid, propionic acid, isobutyric acid, isovaleric acid, and valeric acid) after 24-h in-vitro fermentation with T2DM human fecal microflora.

Data were expressed as means ± SDs (n = 8).

Effect of SNP treatment on the metabolomics alterations in T2DM human fecal matter

As the investigation deepens, intestinal floral metabolites are regarded as crucial substances with involvement in metabolism of the body (Yang et al., 2022). PCA and OPLS models were utilized to explore metabolic differences for the optimization of analysis. As shown in Figs. 9A, 9B, all samples had a confidence interval of 95% (Hotelling’s T-squared ellipse), these models performed an overall intergroup separation, and HC, T2DM, HC+SNP, and T2DM+SNP groups showed significant differences. Additionally, permutation tests were undertaken (200 times) for model verification. As demonstrated in Fig. S1, R2Y had a slope greater than 0 and Q2 showed an intercept smaller than 0.05, indicative of the good predictability and the potential to use the model in subsequent analysis with no overfitting (Liu et al., 2022b).

Figure 9 Metabolite profile after 24-h in-vitro fermentation with T2DM human fecal microflora.

(A) Score plots of PCA; (B) OPLS models; differential metabolite screening in (C) T2DM group vs. HC group, (D) HC+SNP group vs. HC group, and (E) T2DM+SNP group vs. T2DM group; (F) metabolic pathway analysis.

Figures 9C–9E present the screening results for the differential metabolites. In the figure, each point indicates a metabolite. Variable importance projection (VIP) values in the OPLS-DA model are represented by the size of scatter points. The final screening results are indicated by scatter colors, with red, blue, and grey separately represent significantly up-regulated, significantly down-regulated, and non-significantly different metabolites. VIP > 1 (OPLS) and P < 0.05 (t-test) were used to identify significantly distinguishing metabolites. Table 3 displays that the levels of L-leucine, L-alanine, L-valine L-isoleucine, and xylitol in the T2DM group were remarkably higher in comparison to those in the HC group, while the T2DM group contained a significantly lower content of 5-aminovaleric acid than the HC group, which is consistent with previous research (Dash et al., 2023). These metabolites were reversed by T2DM+SNP statistically (vs. T2DM group). As shown in Fig. 9F, the aforementioned differential metabolites were enriched in glutathione metabolism, alanine metabolism, glutamate metabolism, arginine and proline metabolism, urea cycle, glycine and serine metabolism, valine, leucine and isoleucine degradation, glucose-alanine cycle, propanoate metabolism, spermidine and spermine biosynthesis, ammonia recycling, methionine metabolism, and malate-aspartate shuttle, indicating SNP had the potential to regulate the gut metabolites especially amino acid dysbiosis.

Table 3 Identification of differential metabolites.

No.	Metabolites	VIP value	HMDB ID	KEGG ID	T2DM group vs. HC group	
1	L-Leucinea	4.13258	HMDB0000687	C00123	↑	
2	5-Aminovaleric acida	4.03887	HMDB0003355	C00431	↓	
3	L-5-Oxoprolinea	3.31003	HMDB0000267	C01879	↑	
4	L-Valinea	2.74051	HMDB0000883	C00183	↑	
5	L-Alaninea	2.68642	HMDB0000161	C00041	↑	
6	L-Isoleucinea	2.26293	HMDB0000172	C00407	↑	
7	Benzeneacetic acida	2.23157	HMDB0000209	C07086	↓	
8	Xylitola	2.17252	HMDB0002917	C00379	↑	
No.	Metabolites	VIP value	HMDB ID	KEGG ID	T2DM+SNP group vs. T2DM group	
1	Glycerolb	3.69435	HMDB0000131	C00116	↑	
2	L-Leucineb	3.08357	HMDB0000687	C00123	↓	
3	L-Alanineb	2.53218	HMDB0000161	C00041	↓	
4	L-Prolineb	2.4122	HMDB0000162	C00148	↓	
5	L-Isoleucineb	2.31621	HMDB0000172	C00407	↓	
6	Xylitolb	2.19568	HMDB0002917	C00379	↓	
7	Lactic Acidb	2.17976	HMDB0000190	C00186	↑	
8	L-Valineb	1.98641	HMDB0000883	C00183	↓	
9	4-Aminobutanoic acidb	1.8998	HMDB0000112	C00334	↑	
10	Putrescineb	1.86897	HMDB0001414	C00134	↓	
11	DL-Phenylalanineb	1.77224	HMDB0000159	C00079	↓	
12	L-Glutamic acidb	1.36991	HMDB0000148	C00025	↓	
13	Hydracrylic acidb	1.35375	HMDB0000700	C01013	↑	
14	L-Ornithineb	1.31396	HMDB0000214	C00077	↓	
15	Cadaverineb	1.25363	HMDB0002322	C01672	↓	
16	Pyroglutamic acidb	1.19465	HMDB0000267	C01879	↓	
17	1, 3-Propanediolb	1.18227	NA	NA	↓	
18	Glycineb	1.14927	HMDB0000123	C00037	↓	
19	Citric acidb	1.04406	HMDB0000094	C00158	↑	
20	5-Aminovaleric acidb	1.04381	HMDB0003355	C00431	↓	
No.	Metabolites	VIP value	HMDB ID	KEGG ID	HC+SNP group vs. HC group	
1	4-Aminobutanoic acidc	3.96592	HMDB0000112	C00334	↑	
2	Glycerolc	3.55011	HMDB0000131	C00116	↑	
3	L-5-Oxoprolinec	2.8964	HMDB0000267	C01879	↑	
4	L-Alaninec	2.08851	HMDB0000161	C00041	↑	
5	L-Threoninec	2.08373	HMDB0000167	C00188	↑	
6	5-Aminovaleric acidc	1.99475	HMDB0003355	C00431	↓	
7	L-Leucinec	1.86169	HMDB0000687	C00123	↑	
8	Myo-Inositolc	1.81	HMDB0000211	C00137	↑	
9	L-Valinec	1.77472	HMDB0000883	C00183	↑	
10	Benzeneacetic acidc	1.65469	HMDB0000209	C07086	↓	
11	Putrescinec	1.63883	HMDB0001414	C00134	↓	
12	Glycinec	1.63448	HMDB0000123	C00037	↑	
13	L-Prolinec	1.6323	HMDB0000162	C00148	↓	
14	L-Isoleucinec	1.57176	HMDB0000172	C00407	↑	
15	Lactic acidc	1.47344	HMDB0000190	C00186	↑	
16	L-Cysteinec	1.35996	HMDB0000574	C00097	↑	
17	L-Serinec	1.34786	HMDB0000187	C00065	↑	
18	Cadaverinec	1.32952	HMDB0002322	C01672	↓	
19	Citric acidc	1.27905	HMDB0000094	C00158	↑	
20	L-Ornithinec	1.23117	HMDB0000214	C00077	↑	
21	D-Gluconic acidc	1.16057	HMDB0000625	C00257	↑	
22	Xylitolc	1.12604	HMDB0002917	C00379	↑	
23	L-Lysinec	1.05495	HMDB0000182	C00047	↑	
Notes:

a T2DM group vs. HC group.

b T2DM+SNP group vs. T2DM group.

c HC+SNP group vs. HC group.

The further to explore the associations between changed metabolites in human fecal samples and SNP, up-regulated or down-regulated metabolites both in the T2DM+SNP group (vs. T2DM group) and HC+SNP (vs. HC group) were selected (Table 3). The results showed that glycerol, lactic acid, and citric acid were up-regulated after treatment of SNP, moreover, 5-aminovaleric acid, L-proline, putrescine, and cadaverine were down-regulated after SNP treatment.

Analysis of the correlation of metabolites in human fecal matter with intestinal flora

To further explore correlations of metabolites in human fecal matter and intestinal flora after SNP treatment, thermography was used to analyze contents of metabolites showing significant differences with 30 most abundant genera. Figure 10 shows the results. Bifidobacterium was positively associated with glycerol, CO2, 4-aminobutanoic acid, L-cysteine, and D-gluconic acid, and could be negatively correlated with DL-phenylalanine. Faecalibacterium was positively associated with pentanoic acid and 4-aminobutanoic acid, while were negatively correlated with hydracrylic acid. Lactobacillus was positively correlated with xylitol and showed a negative correlation with 5-aminovaleric acid. Dorea was positively associated with phenylalanine, butyric acid, and isovaleric acid, while showed a negative correlation with hydracrylic acid, CH4, myo-inositol, L-threonine, glycerol, CO2, L-cysteine, and D-gluconic acid. Escherichia-Shigella could be positively associated with H2S, L-proline, cadaverine, DL-phenylalanine, benzeneacetic acid, 1,3-propanediol, and were negatively correlated with glycerol, CO2, 4-aminobutanoic acid, L-cysteine, D-gluconic acid, propionic acid, and L-lysine. Phascolarctobacterium was positively associated with 1,3-propanediol, L-proline, cadaverine, and DL-phenylalanine, and was negatively correlated with glycerol, CO2, 4-aminobutanoic acid, L-cysteine, and myo-inositol. Lachnospiraceae were positively associated with 1, 3-propanediol and L-proline, and were negatively correlated with hydracrylic acid and CH4. Parabacteroides had positive correlations with L-leucine, L-valine, and L-isoleucine, while negatively correlated with glycerol, CO2, 4-aminobutanoic acid, L-cysteine, D-gluconic acid, and propionic acid.

Figure 10 Correlation analysis of characteristic metabolites and bacteria.

Corr represents correlation. Red and blue separately denote positive and negative correlations. The darker the color, the stronger the correlation. *P < 0.05, **P < 0.01, and ***P < 0.001.

Discussion

S. ningpoensis is a traditional Chinese medicine that has been utilized for an extended period in the treatment of T2DM (Yan et al., 2022). In T2DM rats, S. ningpoensis polysaccharide (SNP) can enhance the metabolism of lipid and glucose (Ma et al., 2019). Although previous research has shown that SNP is anti-diabetic, the potential effect mechanisms of SNP on gut microbiota composition and metabolism in T2DM patient remain to be clarified. On the basis of the aforementioned theoretical background, SNP probably shows a probiotic effect and thus improves T2DM through regulation of composition structure and metabolite profile of gut microbiota. In the present study, we extracted crude polysaccharide from S. ningpoensis and conducted in-vitro fermentation to detect differences in gut microbiota and metabolites associated with T2DM after treatment of SNP.

Our results found that SNP is predominantly indigestible in the stomach and small intestine, which enhances its interaction and utilization by gut microbiota. SNP significantly reduced microbial community diversity, probably due to the insufficient carbohydrates present during in-vitro fermentation, which restricted the proliferation of gut microbiota (Wu et al., 2023). It is widely acknowledged that T2DM is associated with a low ratio of F/B (Fassatoui et al., 2019; Hamasaki-Matos et al., 2021). The increase in the F/B ratio observed in SNP indicates its potential therapeutic relevance in the management of T2DM.

In addition, SNP selective modulation of the growth of specific bacterial genera has the potential to be beneficial for host health, such as Parabacteroides, Faecalibacterium, Lachnospira. Butyricicoccus, Bifidobacterium, and Lactobacillus. Phascolarctobacterium is also favorable to the human body considering the metabolic characteristics of succinic acid and it also represents a new biomarker in the early diagnosis of T2DM (Li, Hu & Xiong, 2023). Furthermore, Lachnospiraceae, Butyricicoccus, Parabacteroides, Bifidobacterium, and Lactobacillus can ferment plant polysaccharides to produce SCFAs, which exert anti-inflammatory and immune-enhancing properties (Ragavan & Hemalatha, 2024; Wei et al., 2018; Wu et al., 2019b). Seemingly, Bifidobacterium is a genus with the highest consistent literature support and including microbes with a potential protection effect against T2DM. Almost all related reports suggest that this genus is negatively associated with T2DM (Gao et al., 2018; Gurung et al., 2020). Bifidobacterium is able to enhance synthesis of glycogens and down-regulate expressions of genes related to hepatic gluconeogenesis (Kim et al., 2014). Among potentially probiotic bacteria, the highly diverse genus Lactobacillus includes most OTUs in the human guts. Patient-control cross-sectional research shows that abundance of Lactobacillus is positively associated with that of T2DM (Candela et al., 2016; Ni et al., 2018). Researchers reported that Bifidobacterium and Lactobacillus may operate synergistically for improvement of symptoms related to human T2D (Gurung et al., 2020). Lactobacillus and Bifidobacterium generate bile salt hydrolases that enable conversion of primary conjugated bile salts into deconjugated bile acids (BAs), which are then transformed into secondary BAs. Secondary BAs allow activation of the membrane bile acid receptor (TGR5), thus inducing production of GLP-1 (Allin, Nielsen & Pedersen, 2015). SNP inhibited the excessive production of harmful bacteria. The influence of pathogenic Escherichia-Shigella on the host is significant. These pathogens affect the host through various virulence factors, including enterotoxins, Shigella-like toxins, and adhesin fimbriae. Additionally, they can cause damage to the intestinal mucosa, disrupt the actin cytoskeleton of the host cell, and stimulate the secretion of pro-inflammatory cytokines, which can in turn modify the expression of intestinal tight junction proteins (Zhang et al., 2022a). As gram-negative bacteria, Escherichia-Shigella and Desulfovibrionaceae are significantly related to the inflammation, LPS level, IR, and T2DM (Pedersen et al., 2016; Thingholm et al., 2019; Wu et al., 2019a). SNP supplements suppressed abundances of Escherichia-Shigella and Desulfovibrionaceae, in which bacteria served as conditional pathogens in the previous statement.

The elevated level of pro-inflammatory cytokines is a significant factor in T2DM. Gut microbiota enhanced release of inflammatory cytokines through the production of LPS. LPS can promote low-grade inflammation and endotoxemia. Conversely, certain microbiota are recognized to cause increased levels of anti-inflammatory cytokines, such as IL-22 and IL-10, which can enhance the insulin sensitivity, particularly those from the Bacteroides and Lactobacillus (He et al., 2023; Qiu et al., 2017). Lactobacillus has been shown to decrease the effects of LPS-related pathology, particularly by diminishing β-cell dysfunction (Tian et al., 2016). Prior study indicated that Faecalibacterium is negatively correlated with the IL-6 level, likely attributable to its production of butyrate, which serves to inhibit the activation of NF-B (Di Sabatino et al., 2005). In summary, SNP can modulate the diversity and structure of gut microbiota, and increase abundances of beneficial bacteria while decreasing those of pathogenic bacteria. This is the probably the underlying mechanism of hypoglycemia, enhancing insulin resistance and inflammatory responses.

As SNP cannot be digested in the human small intestine, they are able to reach the colon and promote selective growth of beneficial bacteria. These bacteria can generate metabolites conducive to human body, of which the most significant are gas and SCFAs (Tsukuda et al., 2021). Microbiota and chemical interactions in the distal small intestine and colon enable the production of CO2, CH4, H2, and H2S. H2 and CO2 are released during carbohydrate fermentation, while CH4 and H2S are generated by metabolism of H2 and CO2 under the action of methanogens and sulphate reducers (Zhao et al., 2023). The level of total gas production, CO2, CH4, H2, and H2S reduced in the T2DM group (vs. HC group), which was reversed by SNP (T2DM + SNP group vs. T2DM group). The findings are consistent with previous research suggesting that the concentration of H2 in exhaled air is lower in T2DM compared to the healthy group (Misnikova et al., 2024). Prior study indicated that molecular hydrogen can reduce the content of fasting blood glucose, enhance the insulin sensitivity and improve the hepatic glycogen synthesis. Moreover, H2 can reduce the pathological changes of kidney and pancreatic islets associated with T2DM (Ming et al., 2020). H2S reduced fasting blood glucose level, increased insulin sensitivity, and improved glucose tolerance, accompanied by enhanced phosphorylation of Akt and PI3K in muscle (Xue et al., 2013). The volume of gas produced affects the expansion of colonic walls and in turn influences the material transition rate via the colon (Mutuyemungu et al., 2023). Moreover, the great discrepancy between individuals in terms of the gas production pattern is found to be related both to the characteristics of gut microbiota and the available substrates (Kalantar-Zadeh et al., 2019).

SCFAs, also called volatile fatty acids, are organic acids that contain between one and six carbon atoms. They are main metabolites attained by bacterial fermentation of prebiotic polysaccharides in the gut and vital for the health of human intestine (Bai et al., 2023). SCFAs absorbed in the cecum and colon, contributing for about 5–10% to the energy of host supply. They can enter systemic circulation through the portal vein, thereby affecting hepatic and peripheral tissues by directly regulating metabolism or tissue functionality (Fagundes et al., 2024). SCFAs not only serve as essential energy substrates but also play an important role as signaling molecules (Liu et al., 2024a). They have significant physiological effects on various organs, including the pancreas and liver, through multiple complementary pathways. This action enhances insulin sensitivity and glucose metabolism on different levels and influences the progression of T2DM (Pham et al., 2024; Xie et al., 2025).

The levels of total SCFAs (acetic acid, propionic acid, butyric acid, isobutyric acid, and valeric acid) reduced inT2DM group (vs. HC group), which were reversed by SNP (T2DM + SNP group vs. T2DM group), thereby corroborating the findings of prior study such that the levels of total SCFA, acetic acid and butyric acid reduced in T2DM and pre-diabetic states (Wu et al., 2020; Yang et al., 2024). The phenomenon may be explained by a significant increase in the Firmicutes, which can degrade polysaccharides to oligosaccharides and monosaccharides. The majority of butyric acid producers belong to the Clostridium cluster within the phylum Firmicutes (Singh et al., 2022). SNP significantly increased the level of SCFAs, likely due to the enrichment of the Dorea species. Our data showed that the Dorea was positively correlated with butyric acid. The levels of butyric acid and the butyrate-producing bacteria in the gut frequently decreased in pre-diabetes and T2DM. Reversal of this trend can improve disturbances in glycometabolism (Zhang et al., 2022a). Butyric acid, an anti-inflammatory metabolite, is acknowledged for its role in inhibiting pathways that drive the production of pro-inflammatory cytokines (Hodgkinson et al., 2023). It also rescues palmitate-induced insulin resistance by enhancing insulin signaling (Rios-Morales et al., 2022). Additionally, butyric acid has been demonstrated to limit the translocation of LPS in the intestines, thereby alleviating LPS-related effects (Bakshi & Mishra, 2025). In contrast to butyric acid, acetic acid has been associated with increased insulin resistance, enhanced insulin secretion, and elevated ghrelin secretion in response to glucose. These effects are primarily mediated through the activation of the parasympathetic nervous system (Petersen et al., 2019). SCFAs enhance metabolic functions of T2DM through FFAR3 (or GPR41) and FFAR2/GPR43 (e.g., controlling the blood glucose level, as well as IR and GLP-1 secretion) (Puddu et al., 2014). GPR43 shows particular responses to nutrients derived from intestinal flora (such as butyrate, propionate, and acetate) (Yang et al., 2020). SCFAs play a significant role in stimulating the secretion of GLP-1, which increases the expression of hepatic peroxisome proliferator-activated receptor α (PPARα) and activates the AMPK pathway. This activation contributes to promote IR and fat oxidation by inhibiting gluconeogenesis and hepatic adipogenesis (Zhang et al., 2019). Furthermore, GLP-1 interacts with its receptor on pancreatic islet beta cells to facilitate increased insulin secretion (Zaïmia et al., 2023). Besides, SCFAs also as histone deacetylase (HDAC) inhibitors, can up-modulate the expression of metabolism-related genes, such as AMPK, PPARα, and uncoupled protein 1 (UCP-1) by epigenetic modifications (Chen et al., 2024; Zhang et al., 2023).

Gut microbial metabolites were investigated in the present study. SNP have the potential to regulate the gut metabolites especially amino acid dysbiosis. Amino acid has emerged as a novel biomarker for T2DM prediction. Amino acid is the basic unit of protein and the main repository of glucose production and it also shows influences on glucagon and insulin secretion. Reports show that branched chain amino acids (BCAAs, such as leucine, isoleucine, and valine) are signaling molecules that regulate glucose, lipid, and protein metabolism, and have the potential to elevate the risks of T2DM and IR (Lynch & Adams, 2014; Nie et al., 2018). Isoleucine and leucine are insulinotropic, while isoleucine and valine exert gluconeogenic effects. Leucine influences the function of insulin receptors by activating mammalian targets of rapamycin complex 1 (mTORC1) (Ding, Wang & Lu, 2023). The correlation revealed that Parabacteroides was positively correlated with L-leucine, L-valine, and L-isoleucine. Here we found that SNP increased the abundance of Parabacteroides and enhanced BCAA catabolism (Qiao et al., 2022). By profiling responses of the above data to SNP, functions and products of gut microbiota and their influences upon T2DM can be revealed.

Conclusion

Whether SNP can affect the gas volume, metabolite profile, and compositions of gut microbiota during in-vitro fecal fermentation of T2DM patients that can measure gas production, SCFA production, metabolites and 16s rRNA of microbiota responding to SNP addition was investigated. SNP was found to produce larger amounts of CO2 and CH4, and the amount of CO2 produced was highly correlated with Bifidobacterium. SNP also yielded greater amounts of SCFAs, such as acetic acid, propionic acid, butyric acid, isobutyric acid, and valeric acid. Furthermore, SNP played a positive role by modulating structures of gut microbiota and ameliorating metabolic pathways related to dysbiosis of amino acids. The aforementioned results suggest that SNP has potential when seeking to develop active natural functional components for managing diabetes. Integration of quantitative and qualitative analyses of intestinal gases, metabolite and microbiota offers opportunities for improving the understanding of activities and functions of chemical compositions of the gut, physical stimuli and intraluminal bacteria. It also allows elucidation of the significance of fermentation of dietary polysaccharide substrates in diseased and healthy cases.

Several limitations associated with the present study warrant mention: (1) In-vitro fermentation models are regarded as a practical alternative for in-vivo studies for several reasons, such as their being cheaper and quicker. The present study identifies an association between SNP, gut microbiota, metabolites, and T2DM phenotypes by using in-vitro fermentation models, yet it does not establish causality. Further research should incorporate suitable in vivo models to clarify the causal relationships among SNPs, gut microbiota, and therapeutic effects. Future research should focus on validating the feasibility and effectiveness of SNP interventions targeting gut microbiota in T2DM; (2) The microbial composition was detected by 16s rRNA sequencing, which is an extensive and cost-effective technique for measuring microbial diversity, but it can only indirectly estimate microbial function. 16S rRNA short-read sequencing data have limitations in accurately distinguishing microbial species at the species level. Moreover, we were unable to assess the all metabolites by using the GC-MS. The further studies using multi-omics technologies are still needed for revealing molecular mechanisms taking part in hypoglycemic activities of SNP; (3) Polysaccharide in our study were extracted from S. ningpoensis which is natural TCM and health food source, therefore SNPs are comparably safe. However, the potential toxicity of SNP has not been thoroughly investigated. The effective dose of SNP has not yet been determined, and follow-up studies should further explore the individualized dose and safety assessment. The composition molecular weight, and bioactivity, of polysaccharide warrants further investigation. These investigations will enhance the translation of experimental findings into clinical applications.

Supplemental Information

Supplemental Information 1 Preparation of simulated digestion solution.

Supplemental Information 2 Percent of community abundance on phylum level.

Supplemental Information 3 Percent of community abundance on genus level.

Supplemental Information 4 Statistical validation of the corresponding PLS-DA model using two hundred permutations analysis.

(A) T2DM group vs. HC group; (B) HC+SNP group vs. HC group; (C) T2DM+SNP group vs. T2DM group.

Supplemental Information 5 Raw data of gas composition and concentration.

Supplemental Information 6 Raw data of SCFA composition and concentration.

Supplemental Information 7 Raw data of metabolite composition and concentration.

Additional Information and Declarations

Competing Interests

The authors declare that they have no competing interests.

Author Contributions

Yang Zhao analyzed the data, prepared figures and/or tables, and approved the final draft.

Juwei Wen analyzed the data, prepared figures and/or tables, and approved the final draft.

Yu Yang analyzed the data, prepared figures and/or tables, and approved the final draft.

Lina Jia performed the experiments, analyzed the data, prepared figures and/or tables, and approved the final draft.

Qian Ma performed the experiments, prepared figures and/or tables, and approved the final draft.

Weiguo Jia conceived and designed the experiments, authored or reviewed drafts of the article, and approved the final draft.

Wei Qi conceived and designed the experiments, prepared figures and/or tables, authored or reviewed drafts of the article, and approved the final draft.

Human Ethics

The following information was supplied relating to ethical approvals (i.e., approving body and any reference numbers):

Biomedical Ethics Committee of West China Hospital of Sichuan University (protocol number 2018 (286)).

DNA Deposition

The following information was supplied regarding the deposition of DNA sequences:

The 16s rRNA sequences are available at SRA: PRJNA1191790.

Data Availability

The following information was supplied regarding data availability:

The raw measurements are available in the Supplemental Files.

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
