# Peer review of "In vitro fermentation characteristics of polysaccharide from Scrophularia ningpoensis and its effects on type 2 diabetes mellitus gut microbiota"

_PeerJ, doi:10.7717/peerj.19374_

## Round 0.1 · original submission · Major Revisions

Dear authors, I ask you to carefully correct the shortcomings pointed out by the reviewers.

Reviewer 1 ·

Basic reporting

No Comment

Experimental design

No Comment

Validity of the findings

No Comment

Additional comments

The manuscript is well displayed, written and discussed, the results are promising, indeed, there are some comments for improvement: -
1. The introduction should include more details, and clearly state the aim of the study, and the necessity of interventions in T2DM that target gut microbiota.
2. In results and discussion, it would be better if the findings are compared with the previous studies.
3. Language and Grammar, the manuscript contains several grammatical and syntax errors, English editing is necessary through the whole manuscript.

·

Basic reporting

I recommend accept with changes that include: Remove Corres, from author name Wei Qi and put symbol instead.. Add other important new anti-diabetic classes of drugs ( Incretin mimmetics and SGLT2 inhibitors) with their brief mechanisms of action in the introduction.Provide title for each figue to make them understandable by the readers.

Experimental design

Cite the inclusion and exclusion criteria. Cite the diagnostic criteria of type 2 diabetes used in the study.

Validity of the findings

The results are novel and sufficient details are provided that maes the study replicable. The data is plausible and credible. Statistical power is adequate. The conclusion correlates with results.
The author is advised to discuss the partially known biomolecular mechanisms of LPS of SNP such as repairing the gut barrier, reshaping gut microbiota, changing metabolites,regulation of ant-inflammatory activity and immune function. Discuss LPS regulatory role in signaling pathways. Discuss the molecular mechanisms of SNP to improve insulin sensitivity. Discuss the limited resolution and accuracy of used methods including 16s rRNA,sequencing, metagenomics and chromatography- Mass spectroscopy. Discuss the safety and efficacy of the extract. Add limitations of the study. Modify reference No.22 (many Authors) as per journal refrerevce styles guidelines.

Reviewer 3 ·

Basic reporting

The manuscript is generally written in clear and professional English; however, some sentences are overly complex and repetitive. Simplifying these sentences would improve readability.
Some major grammatical and stylistic errors are present.
The authors highlight the limited research on SNP and its effects, particularly in the context of diabetes, which adds novelty to this study.
The manuscript is well-structured, with clear sections for introduction, methods, and results. Add a discussion section to interpret the findings, highlight the study's novelty, and compare the results with existing literature.

Experimental design

The use of an in-vitro model is an inherent limitation of the study. This should be acknowledged, and the need for in-vivo validation should be highlighted.
Future studies should be proposed to evaluate the effects of SNP more comprehensively. For example: For ex.; "How can SNP contribute to the management of T2DM?"
"What is the clinical significance of the increase in SCFA production?"
Is the production of SCFAs in T2DM consistent with findings from other studies?

Validity of the findings

The role of increased SCFA production (e.g., butyrate, acetate) in improving glucose metabolism and reducing inflammation should be further discussed.
The significance of bacterial changes, such as the increase in Bifidobacterium and the decrease in Escherichia-Shigella, should be better contextualized.

Additional comments

The comparison of findings with the literature is insufficient or superficial in some parts. The following questions should be addressed:
How have the effects of SNP been investigated previously? What novelty does this study provide?
Is the production of SCFAs in T2DM consistent with findings from other studies? If there are discrepancies, what are the possible reasons?

In Figure 5B, due to the large number of bacteria, similar colors have been used, making the visualization difficult to interpret. Would it be possible to upload the article in an interactive format, where users can hover over a bacterium with their mouse to see its name and abundance value?

In Table 2, comparisons have been made, and increase/decrease arrows are included. However, the table does not immediately show which groups are being compared. The comparison method should be specified, at least in the table caption or through a brief explanation

---

## Round 0.2 · accepted · Accept

Dear Dr. Qi, I am pleased to inform you that your article has been accepted for publication in our journal. We look forward to your next manuscripts.